# Tribological Performance and Application of Antigorite as Lubrication Materials

**Zhimin Bai [1,\*], Guijin Li [2], Fuyan Zhao [3] and Helong Yu [4]**

[1] Beijing Key Laboratory of Materials Utilization of Nonmetallic Minerals and Solid Wastes, University of Geosciences, Beijing 100083, China
[2] Shandong Branch, China Building Materials Academy, Beijing 100024, China; liguijin007@126.com
[3] Qingdao Center, Lanzhou Institute of Chemical Physics, Lanzhou 730000, China; fyzhao@licp.cas.cn
[4] National Key Laboratory for Remanufacturing, Army Academy of Armored Forces, Beijing 100072, China; helong.yu@163.com
[\*] Correspondence: zhimibai@cugb.edu.cn

**Abstract:** Antigorite is a Mg-rich 1:1 trioctahedral-structured layered silicate mineral. In recent decades, many studies have been devoted to investigating the tribological performance and application of antigorite as lubrication materials. This article provides an overview of the mineralogy, thermal decomposition and surface modifications of antigorite powders, as well as the recent advancement that has been achieved in using antigorite to reduce friction and wear of friction pairs. The tribological performance of antigorite powders and its calcined product in different lubricating media, such as oil, grease and solid composites have been comprehensively reviewed. The physico-chemical characteristics of surface layers of the friction pairs are discussed. Applications and mechanisms of lubricity and anti-wear of antigorite are highlighted.

**Keywords:** antigorite; tribological performance; lubricity; application

## 1. Introduction

Antigorite is a Mg-rich 1:1 trioctahedral-structured layered silicate mineral of the serpentine group. Layered minerals are known as laminar structure minerals or layer lattice minerals, have layer crystal structure in which the atoms within a layer are held together by a chemical bond stronger than the bond between the layers. This provides an isotropic shear property with preferred easy shear parallel to the basal planes of the crystallites, and result in the ability of basal planes to slide easily over one another [1]. This may be the reason that layered minerals can provide low friction coefficients and be used as solid lubricants [2–4].

The earlier researches on frictional behavior of antigorite were conducted by geologists and geophysicists, with the intent of seeking information about the strength and sliding stability of natural faults containing antigorite [5–11]. In the recent decades, there has been considerable interest in using antigorite to reduce friction and wear of machinery and equipment. Significant progress has been achieved in the research and development of antigorite's application in the lubrication of machine components. The research results indicated that antigorite micro–nano powder (AMNP) may significantly reduce friction coefficient and wear of friction pairs, and be one kind of excellent anti-frictional and lubrication materials [12–27]. This paper reviews the mineralogy, thermal decomposition, lubrication behavior and application of antigorite lubricant. The main goal is to draw attention of researchers to antigorite and to assist the researchers in better understanding the function of the antigorite lubricant and its lubrication mechanisms.

## 2. Mineralogy and Powder Characteristics of Antigorite

The theoretical chemical composition of antigorite is $SiO_2$ 43.37%, MgO 43.63%, $H_2O$ 13.00%, with an ideal formula of $Mg_6Si_4O_{10}(OH)_8$. They can exhibit the substitution of $Si^{4+}$ cations by $Al^{3+}$, $Fe^{3+}$, or $Cr^{3+}$, and $Mg^{2+}$ by $Fe^{2+}$, $Mn^{2+}$, $Ca^{2+}$, or $Ni^{2+}$ [28,29]. The actual composition of naturally occurring antigorites exhibits variation in the $SiO_2$, MgO and $H_2O$, and minor proportion of $Al_2O_3$, FeO, $Fe_2O_3$, MnO, CaO, $Cr_2O_3$ and NiO are also present [4,30–32].

Antigorite belongs to the group of trioctahedral 1:1 layered silicates, consisting of one tetrahedral (T) and one octahedral (O) sheet. The T sheet is formed by the two-dimensional polymerization of Si-centered tetrahedra sharing three out of four oxygen atoms with other tetrahedra. The unshared oxygen atoms are bonded to Mg atoms that jointly with OH groups form the octahedral sheets (O sheet). The T- and O- sheets are variably arranged and stacked one above another, and the thickness of the T-sheet is thinner than that of the O- sheet, resulting is a subtle dimensional misfit between the two. This misfit can be reduced by substitutions in the tetrahedron, which in turn reduces interlayer strain and consequently enhances the mineral's stability [28,33]. The layers in the structure are linked by hydrogen bonds that form by pairing oxygen on the basal tetrahedral surface of one layer with an OH-group on the upper octahedral surface of the layer below. These bonds are generally long and weak but can be modified by the degree of substitution [33]. This crystal structure gives rise to typically platy or lamellar along (001) and perfect cleavages on {001} of antigorite.

Mohs hardness of antigorite is between 2.5 to 3.5, the Vickers microhardness are within a range from 196.3 to 204.9 [4]. Its theoretical and measured densities are 2.61 $g/cm^3$ and about 2.65 $g/cm^3$, respectively [34].

Figure 1 shows a typical DTA/TG thermogram (Differential Thermal Analysis/Thermogravimetric Analysis) of the antigorite powders. The DTA curve shows two endothermic peaks and two exothermic peaks. The first broad endothermic peak in the 70–107 °C is due to the loss of adsorbed water, the second intense endothermic peak in the 620–707 °C may be attributed to the loss of structural water of antigorite. The first apparent exothermic peak at 793 °C partially overlaps the 831 °C sharp endothermic peak (the second), which indicates the formation of the new minerals. The former corresponds to forsterite formation, the latter corresponds to the formation of enstatite + forsterite [4,35,36]. The TG curve shows outstanding weight losses during 620–707 °C, which corresponds to the intense endothermic peak in the DTA curves. Apparent activation energy of the reaction in the temperature range 612–708 °C was 255 kJ/mol for antigorite [37].

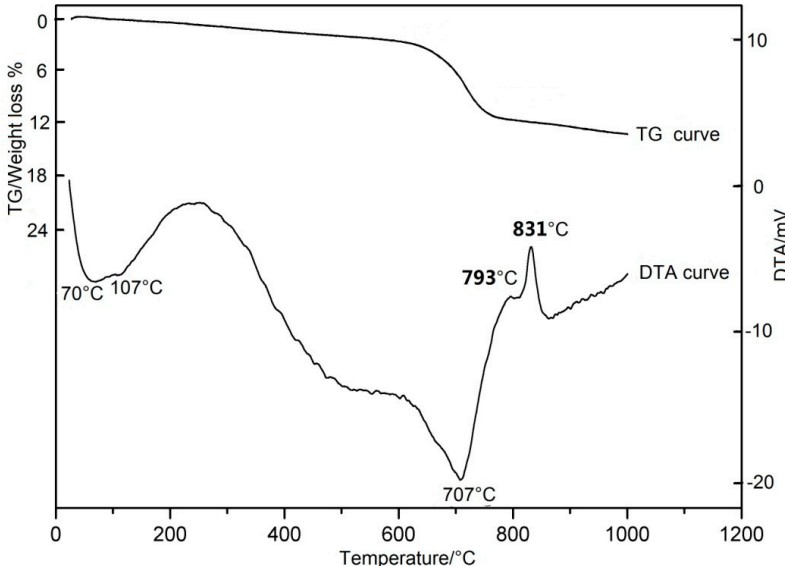

**Figure 1.** DTA/TG thermogram of the antigorite [4].

Natural antigorite is commonly bladed or fibrous. The antigorite as a lubrication material is usually bladed or lamellar (Figure 2), and processed into micro-nano scale powders by powder processing equipment [4,21–23].

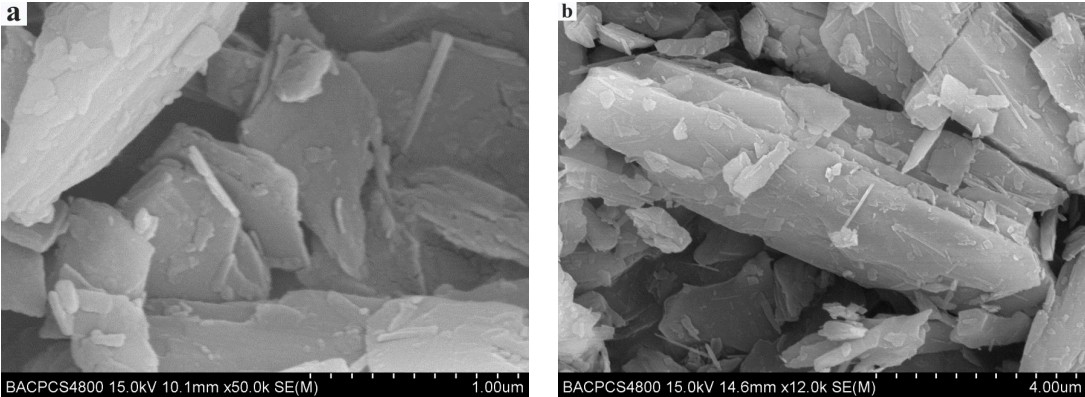

**Figure 2.** Bladed and lamellar antigorite [4]. (**a**,**b**) are different particles in the same sample.

Antigorite showed a positive zeta potential over a wide pH range with a pH value of close to 10 at its isoelectric point [38], which is readily wetted by water and is said to be hydrophilic [39–41]. In order to improve the compatibility of AMNP and organic lubrication medium, the surface modifications of AMNP have been investigated [19,21,22]. The formation of organically modified antigorite is carried out by applying the surfactants as boric acid ester, sorbitan monostearate (Span60), polysorbate 60 (Tween60), oleic acid, polyvinyl pyrrolidone (PVP), silane coupling agent (HK560), sodium dodecyl benzene sulfonate, trolamine, stearic acid, sodium stearate [19,21,22,42–45]. The studies investigating the surface modifications of AMNP have shown that modified powders present superior dispersion and suspension stability in the nonpolar solvent. It was found that the modifiers of oleic acid or Span60 combined with borate at 1:1 (mass ratio) had the best modification effect on AMNP [44].

## 3. Tribological Performances of Antigorite

The key research contents of tribological performances of antigorite are shown in Figure 3. The antigorite powders used for friction and wear experiments have been processed into micron (average particle size 1–20 μm), or sub-micron (average particle size 100 nm to 1 μm) or nanometer scales (<100 nm). The antigorite micro-nano powder (AMNP) used for friction and wear experiments was mixed with different types of lubricating oil to form a suspended solution (Oil + AMNP system, such as Table 1 (No.1 to 46); or mixed with matrix raw materials to make solid composite materials, such as Table 1 (No.47 to 66); or evenly mixed with base grease to make a complex grease, such as Table 1 (No.67 to 74).

**Table 1.** List of tribological performance of antigorite powder.

| | Friction and Wear Experiments | | | |
|---|---|---|---|---|
| No. | System and Test Devices | $\mu_c$ (%) | $wear_c$ (%) | References |
| (1) | Oil + AMNP 0.86 μm (1.0%) (FB) | 19.3 | 16.41 | [4] |
| (2) | Oil + AMNP(FB) | - | 18.7 | [13] |
| (3) | Oil + AMNP < 10.0 μm (0.025%) (FB) | - | 20.5 | [14] |
| (4) | Oil + AMNP < 0.5 μm (0.5%) (P-D) | 68.1 | - | [18] |
| (5) | Oil + AMNP < 0.5 μm (0.5%) (P-F) | 21.3 | 49.7 | [19] |
| (6) | Oil + AMNP < 1.0 μm (1.5%) (R-D) | 55.3 | 82.0 | [22] |
| (7) | Oil + AMNP < 0.8 μm (1.0%) (FB) | 21.7 | - | [24] |
| (8) | Oil + AMNP 3.0 μm (1.0%) (P-D) | 9.8 | 23.3 | [26] |
| (9) | Oil + AMNP 3.0 μm (0.1%) (P-D) | - | 30.4 | [27] |
| (10) | Oil + AMNP 1.0 μm (1.5%) (P-D) | 58.6 | 61.4 | [46] |
| (11) | Oil + AMNP (2.5%) (D-D) | 10.0 | 50.0 | [47] |
| (12) | Oil + AMNP < 1.0 μm (1.5%) (P-F) | 29.0 | 18.0 | [48] |
| (13) | Oil + AMNP 0.3 μm (1.0%) (FB) | 30.8 | 15.7 | [49] |
| (14) | Oil + AMNP < 1.0 μm (0.5%) (P-D) | 41.2 | 28.0 | [50] |
| (15) | Oil + AMNP < 1.0 μm (0.5%) (P-D) | 33.3 | - | [50] |
| (16) | Oil + AMNP < 0.3 μm (10.0%) (FB) | 14.8 | 11.82 | [51] |
| (17) | Oil + AMNP < 0.4 μm (1.0%) (FB) | 14.8 | 11.6 | [51] |
| (18) | Oil + AMNP < 0.5 μm (0.05%) (P-F) | 16.4 | 56.7 | [52] |
| (19) | Oil + AMNP (0.5%) (FB) | 53.3 | 42.6 | [53] |
| (20) | Oil + AMNP < 2.0 μm (0.5%) (P-F) | 18.4 | 42.4 | [54] |
| (21) | Oil + AMNP 0.19 μm (0.25%) (FB) | 18.1 | 32.8 | [55] |
| (22) | Oil + AMNP 1.0 μm (0.5%) (FB) | - | 79.7 | [56] |
| (23) | Oil + AMNP < 0.5 μm (0.5%) (F-C) | 9.7 | 40.7 | [57] |
| (24) | Oil + AMNP < 1.0 μm (1.0%) (R-D) | 50.0 | - | [58] |
| (25) | Oil + AMNP 0.02 μm (2.0%) (P-D) | - | 48.0 | [59] |
| (26) | Oil + AMNP 0.2 μm (3.0%) (TW) | 89.5 | - | [60] |
| (27) | Oil + AMNP < 0.3 μm (0.5%) (P-F) | 12.3 | 66.7 | [61] |
| (28) | Oil + AMNP 0.3 μm (0.5%) (P-F) | 36.6 | 53.6 | [62] |
| (29) | Oil + AMNP < 1.0 μm (1.0%) (F-C) | 68.3 | - | [63] |
| (30) | Oil + AMNP < 0.5 μm (0.5%) (P-F) | 15.5 | 50.0 | [64] |
| (31) | Oil + AMNP 1.6 μm (0.5%) (P-F) | 51.5 | 29.6 | [65] |
| (32) | Oil + AMNP 300 °C (1.5%) (P-F) | 40.0 | 39.0 | [48] |
| (33) | Oil + AMNP 600 °C (1.5%) (P-F) | 38.0 | 23.0 | [48] |

**Table 1.** *Cont.*

| | Friction and Wear Experiments | | | |
|---|---|---|---|---|
| **No.** | **System and Test Devices** | $\mu_c$ (%) | $wear_c$ (%) | **References** |
| (34) | Oil + AMNP 800 °C (1.5%) (P-F) | −8.0 | −2.0 | [48] |
| (35) | Oil + AMNP 1050 °C (1.5%) (P-F) | −13.0 | −8.0 | [48] |
| (36) | Oil + AMNP 200 °C (1.0%) (FB) | 33.9 | 17.1 | [49] |
| (37) | Oil + AMNP 500 °C (1.0%) (FB) | 27.2 | 11.4 | [49] |
| (38) | Oil + AMNP 600 °C (1.0%) (FB) | 27.1 | 11.4 | [49] |
| (39) | Oil + AMNP 800 °C (1.0%) (FB) | 26.6 | 8.6 | [49] |
| (40) | Oil + AMNP + La(OH)$_2$ (0.5%) (FB) | 24.6 | 41.9 | [25] |
| (41) | Oil + AMNP (0.46%) + Cu (0.04%) (P-F) | 31.3 | 65.1 | [19] |
| (42) | Oil + AMNP (0.25%) + Ce (0.25%) (FB) | 43.9 | 50.0 | [54] |
| (43) | Oil + AMNP (0.475%) + La (0.025%) (FB) | 34.2 | 68.8 | [66] |
| (44) | Oil + AMNP (0.07%) + Ni (0.1%) + Cu (0.3%) (FB) | 37.4 | 34.0 | [67] |
| (45) | Oil + AMNP (0.25%) + Mo (0.3%) (FB) | 32.8 | 53.2 | [55] |
| (46) | Oil + AMNP (0.48%) + La (0.02%) (P-F) | 29.1 | 60.0 | [64] |
| (47) | PTFE + AMNP (1%) (RF-RC) | 10.0 | 95.6 | [68] |
| (48) | PTFE + AMNP (2%) (RF-RC) | 15.0 | 99.8 | [68] |
| (49) | PTFE + AMNP (5%) (RF-RC) | 10.0 | 99.6 | [68] |
| (50) | PTFE + AMNP (10%) (RF-RC) | 0.0 | 99.4 | [68] |
| (51) | PTFE + AMNP (10%) (P-D) | 9.5 | - | [69] |
| (52) | PTFE + AMNP (10%) (P-D) | 2.7 | 94.4 | [70] |
| (53) | Cu$_{60}$Zn$_{40}$ + AMNP (1.0%) (P-D) | 11.1 | 120 | [57] |
| (54) | TiAl + AMNP (7.0%) 25 °C (P-D) | 15.0 | 24.6 | [71] |
| (55) | TiAl + AMNP (7.0%) 200 °C (P-D) | 8.8 | 24.3 | [71] |
| (56) | TiAl + AMNP (7.0%) 600 °C (P-D) | 20.8 | 41.9 | [71] |
| (57) | TiAl + AMNP (7.0%) 800 °C (P-D) | 8.0 | 11.4 | [71] |
| (58) | Al$_{88}$Si$_{12}$ + AMNP (3.0%) (P-D) | 8.6 | 32.7 | [72] |
| (59) | NiAl + AMNP (8.0%) 100 °C (P-D) | 8.2 | 40.5 | [62] |
| (60) | NiAl + AMNP (8.0%) 300 °C (P-D) | 20.9 | 53.1 | [62] |
| (61) | NiAl + AMNP (8.0%) 500 °C (P-D) | 39.8 | 62.6 | [62] |
| (62) | NiAl + AMNP (8.0%) 700 °C (P-D) | 36.7 | 58.7 | [62] |
| (63) | NiAl + AMNP (2%) (P-D) | 17.7 | 15.8 | [73] |
| (64) | NiAl + AMNP (5%) (P-D) | 31.5 | 25.3 | [73] |
| (65) | NiAl + AMNP (8%) (P-D) | 45.2 | 29.5 | [73] |
| (66) | NiAl + AMNP (11%) (P-D) | 42.7 | 22.3 | [73] |

**Table 1.** *Cont.*

| Friction and Wear Experiments | | | | | | | |
|---|---|---|---|---|---|---|---|
| **No.** | **System and Test Devices** | | | | $\mu_c$ (%) | *wear*$_c$ (%) | **References** |
| **No.** | **System (four-ball tester)** | $P_{cr}$ | $P_{weld}$ | $d_{wear}$ | $\mu_c$(%) | *wear*$_c$ (%) | **References** |
| (67) | Grease + AMNP (0%) | 549 | 1303 | 0.78 | | | [14] |
| (68) | Grease + AMNP (1%) | 588 | 1470 | 0.62 | | 20.5 | [14] |
| (69) | Grease + AMNP (2%) | | | | 40.5 | 72.0 | [52] |
| (70) | Grease + AMNP (3%) | | | | 7.7 | 7.6 | [74] |
| (71) | Grease + AMNP (0.75%) + Bi (2.25%) | | | | 23.3 | 18.2 | [74] |
| (72) | Grease + AMNP (0%) | 413 | 1232 | 0.73 | | | [75] |
| (73) | Grease + AMNP (0.5%) | 547 | 1565 | 0.58 | | 20.5 | [75] |
| (74) | Grease + AMNP (0.7%) | 547 | 1565 | | | 10.3 | [76] |
| **Comparison of Surface Hardness of Friction Pair** | | | | | | | |
| **No.** | **Experimental condition** | | | | **Without AMNP (GPa)** | **With AMNP (GPa)** | **References** |
| (75) | Load 50 N, 45# steel, Friction time 10 h | | | | 6.27 | 9.37 | [19] |
| (76) | Diesel cylinder after running $16 \times 10^4$ km | | | | 6.26 | 11.37 | [20] |
| (77) | Load 10 N, 1045 steel, Friction time 1 h | | | | 3.5 | 5.0 | [77] |
| (78) | Load 200 N, 45# steel, Friction time 2 h | | | | 3.47 | 6.51 | [52] |
| (79) | Load 11.5 N, TiAl matrix, Friction time 0.5 h | | | | 3.69 | 6.15 | [71] |
| (80) | Load 400 N, 45# steel, Friction time 2 h | | | | 3.85 | 5.22 | [58] |
| (81) | Load 30 N, Tin Bronze, Friction time 1h | | | | 2.4 | 3.5 | [61] |
| (82) | 45# steel, Friction time 160 h | | | | 9.0 | 15.0 | [78] |
| (83) | Load 400 N, 45# steel, Friction time 8 h | | | | 3.81 | 4.96 | [79] |
| (84) | 45# steel, Friction time 1 h | | | | 2.39 | 3.18 | [80] |
| (85) | Cast iron, Friction time 72 h | | | | 10.14 | 11.17 | [81] |
| (86) | Load 38.34 N, 45# steel, Friction time 24 h | | | | 238.8 | 329.9 (Hv Hardness) | [82] |
| (87) | Diesel cylinder after running $29.3 \times 10^4$ km | | | | 524 | 1119 (Hv Hardness) | [83] |
| (88) | Diesel cylinder after running $50 \times 10^4$ km | | | | 540 | 1185 (Hv Hardness) | [59] |

**Table 1.** *Cont.*

| | Comparison of Surface Elastic Modulus of Friction Pairs | | | |
|---|---|---|---|---|
| **No.** | **Experimental condition** | **Without AMNP (GPa)** | **With AMNP (GPa)** | **References** |
| (89) | Load 50 N, 45# steel, Friction time 10 h | 253.9 | 285.2 | [19] |
| (90) | Diesel cylinder after running $16 \times 10^4$ km | 66.5 | 179.0 | [20] |
| (91) | Load 10 N, 1045 steel, Friction time 1 h | 210.0 | 235.0 | [77] |
| (92) | Load 200 N, 45# steel, Friction time 2 h | 214.7 | 236.6 | [52] |
| (93) | Load 400 N, 45# steel, Friction time 2h | 238.9 | 221.2 | [58] |
| (94) | Load 30 N, Tin Bronze, Friction time 1 h | 140.0 | 180.0 | [61] |
| (95) | 45# steel, Friction time 160 h | 200.0 | 370.0 | [78] |
| (96) | Load 400 N, 45# steel, Friction time 8 h | 196.5 | 213.3 | [79] |
| (97) | 45# steel, Friction time 1 h | 250.0 | 212.1 | [80] |
| (98) | Cast iron, Friction time 72 h | 208.0 | 296.0 | [81] |
| | Comparison of Surface Roughness (Ra) of Friction Pairs | | | |
| **No.** | **Experimental condition** | **original surface ($R_a$/μm)** | **friction surface ($R_a$/μm)** | **References** |
| (99) | Load 220 N, sliding speed 0.35 m/s, run time1 h | 0.046 | 0.036 | [25] |
| (100) | Contact stress 0.33–0.7MPa, run time 27 h | 0.536 | 0.386 | [82] |
| (101) | Contact stress 0.33–0.7MPa, run time 27 h | 0.369 | 0.260 | [82] |
| (102) | Diesel cylinder after running $50 \times 10^4$ km | 2.5 | 0.0267 | [59] |
| (103) | Contact stress 7.64 MPa, run time 4 h | 0.742 | 0.207 | [59] |
| (104) | Contact stress 7.64 MPa, run time 4 h | 3.706 | 2.528 | [59] |
| (105) | Contact stress 7.64 MPa, run time 4 h | 1.424 | 1.276 | [59] |
| (106) | Contact stress 5.00 MPa, run time 1 h | 0.636 | 0.280 | [84] |
| 107) | Contact stress 5.00 MPa, run time 1 h | 0.229 | 0.155 | [84] |
| (108) | load400 N, rotary speed 192 r/min, run time1 h | 0.427 | 0.083 | [80] |
| (109) | load750 N, rotary speed 200 r/min, run time100 h | 0.320 | 0.110 | [65] |

**Table 1.** *Cont.*

| | Application of AMNP Lubrication Materials in Industrial Equipment | |
|---|---|---|
| **No.** | **Equipment type and application result** | **References** |
| (110) | Gearing | |
| | average power consumption of driving motor reduced by 5.0% | [4] |
| | temperature of lubrication oil reduced by 9.7% | [4] |
| | average power consumption of driving motor reduced by 13% | [53] |
| | vibration amplitude of reduction gears reduced by 48% | [85] |
| (111) | Air compressor | |
| | average power consumption of driving motor reduced by 4.5% | [4] |
| | temperature of lubrication oil reduced by 15.8% | [4] |
| | consumption of lubrication oil reduced by 94.5% | [59] |
| | average power consumption of driving motor reduced by 9.12% | [59] |
| (112) | Automobile engine | |
| | cylinder burst pressure increased by 3.9% | [86] |
| | fuel consumption of automobile engine reduced by 7.0% | [86] |
| | CO emissions of automobile engine reduced by 39.5% | [86] |
| | CH emissions of automobile engine reduced by 29.5% | [86] |
| | cylinder burst pressure increased by 11% | [59] |
| | fuel consumption of automobile engine reduced by 1.2–6.0% | [59] |
| (113) | Locomotive engine | |
| | diesel consumption of engine reduced by 2.5% | [87] |
| | lubricating oil consumption of engine reduced by 14.3% | [87] |
| | cylinder burst pressure of engine increased by 2.7% | [59] |
| | diesel consumption of engine reduced by 2.2% | [58] |

AMNP -antigorite micro-nano powder; $P_{cr}$ = critical load (N); $P_{weld}$ = welding load (N); $\mu_c$ = ($\mu$ of without AMNP $-$ $\mu$ of with AMNP) $\times$ 100/ $\mu$ of without AMNP; *Wear*$_c$ = (*wear* of without AMNP/*wear* of with AMNP) $\times$ 100/*wear* of without AMNP; $D_{wear}$ = arithmetic mean diameter of wear spots on bottom balls(mm); PTFE = polytetrafluoroethylene; FB = four-ball tester; P-D = pin-on-disk tester; P-F = pin-on-flat tester; R-D = ring-on-disk tester; RF-RC = rectangular flat on rotating cylinder tester; D-D = disk-on-disk tester; TW = thrust washer tester; F-C = flat-on-cylinder tester.

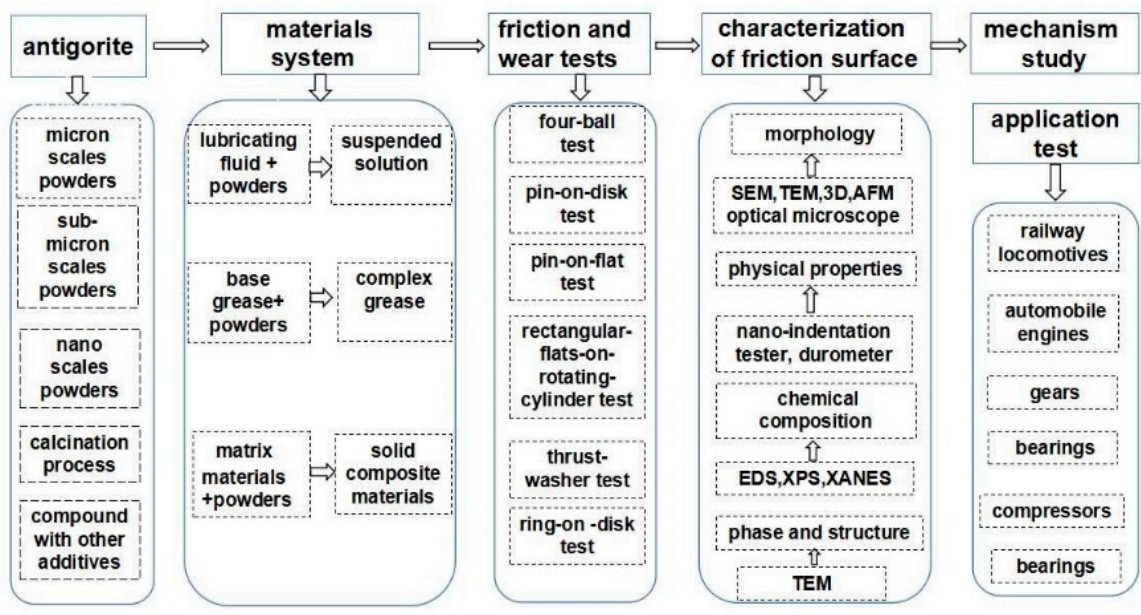

**Figure 3.** Schematic diagram of the tribological performance study.

AMNP-oil system: tests were applied to the various AMNP-oil systems in an effort to determine the tribological performances of antigorite. The principal devices used were the four-ball tester (FB), pin-on-disk tester (P-D) and pin-on-flat tester (P-F); however, ring-on-disk tester (R-D), rectangular flat on rotating cylinder tester (RF-RC), disk-on-disk tester (D-D), thrust washer tester (TW) and flat-on-cylinder tester (F-C) have also been used [12–27]. AMNP used for the study had a particle size less than 3 microns, and most of them were less than 1 micron. The amount of AMNP added to the liquid lubricating medium was <10%, and most of them were <2% (Table 1). The results showed that the friction coefficient (μ) of oil with AMNP was reduced by 9.7–89.5% (Figure 4), the wear decreased by 11.6–82% (Figure 5), the temperature of the lubricating medium reduced by 35.6–44.3% (Table 1, No.1 to 46; Figure 6). It was found that AMNP with an average particle size of less than 2 microns all show the effect of reducing μ, while AMNP with an average particle size of more than 5 microns significantly increase the μ. The general rule is that the smaller the AMNP particle size, the lower the friction coefficient and wear [44]. The effect of AMNP on reducing the friction coefficient and wear can be summarized in the following four aspects [4]. First, the existence of AMNP and mechanical attachment of AMNP to the metal surface could prevent gross metal contact but also shear easily. Secondly, the mechanical spacing effect where AMNP serve as spacers between friction pairs to prevent bare metal-on-metal direct contact. Thirdly, the colloidal with the AMNP effect where the presence of AMNP moves substantial amounts of fluid to the boundary layers, changing the elastic properties of the fluid. Last, the interaction of AMNP with a metallic surface resulted the formation of a smooth transfer film.

Calcined antigorite-oil system: based on the dehydration and phase transition characteristics of antigorite at high temperature [4,35,36], the tribological performances of AMNP calcined at different temperatures were evaluated [24,48,49] (Table 1, No.32 to 39). It was shown that the tribological properties of AMNP under 300 °C were better than those of the uncalcined, those with calcined temperature between 300 °C to 600 °C were similar to those of uncalcined samples, both of which can reduce the friction coefficient and wear of the friction pairs. The effect of reducing friction coefficient and wear of friction pairs still exists in the AMNP with the calcination temperature of 600–800 °C, but it is obviously worse than that of the uncalcined AMNP. In sharp contrast, the friction coefficient and friction pair wear of the calcined AMNP at a temperature higher than 800 °C are higher than that of the base oil and of uncalcined AMNP. It is found that the difference of tribological performances of AMNP calcined at different temperatures is closely related to the high temperature dehydroxylation

and phase transition of antigorite [24,48]. The layered structure of AMNP that calcined below 600 °C is not damaged, and there is no phase transition, so its tribological performance is similar to that of uncalcined AMNP. The AMNP that calcined at 600–800 °C lost its hydroxyl group water and the inter-layer sliding capacity became worse, resulting in the increase of the friction coefficient and wear of the friction pair. It is interesting to note that calcined AMNP at a temperature higher than 800 °C changed into enstatite and forsterite with high hardness and high density [4], which is the internal factor of the significant increase of friction coefficient and friction pair wear.

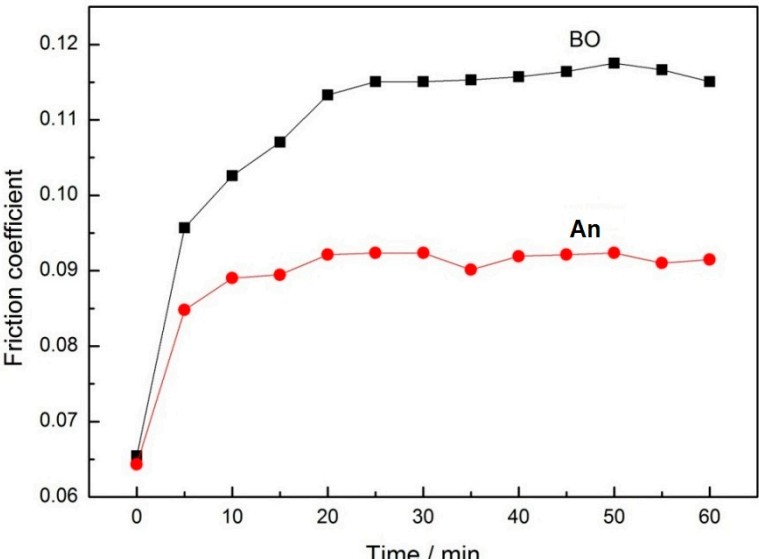

**Figure 4.** Changes of friction coefficient in four-ball friction test [4]. BO—without antigorite micro–nano powder (AMNP); An—with AMNP.

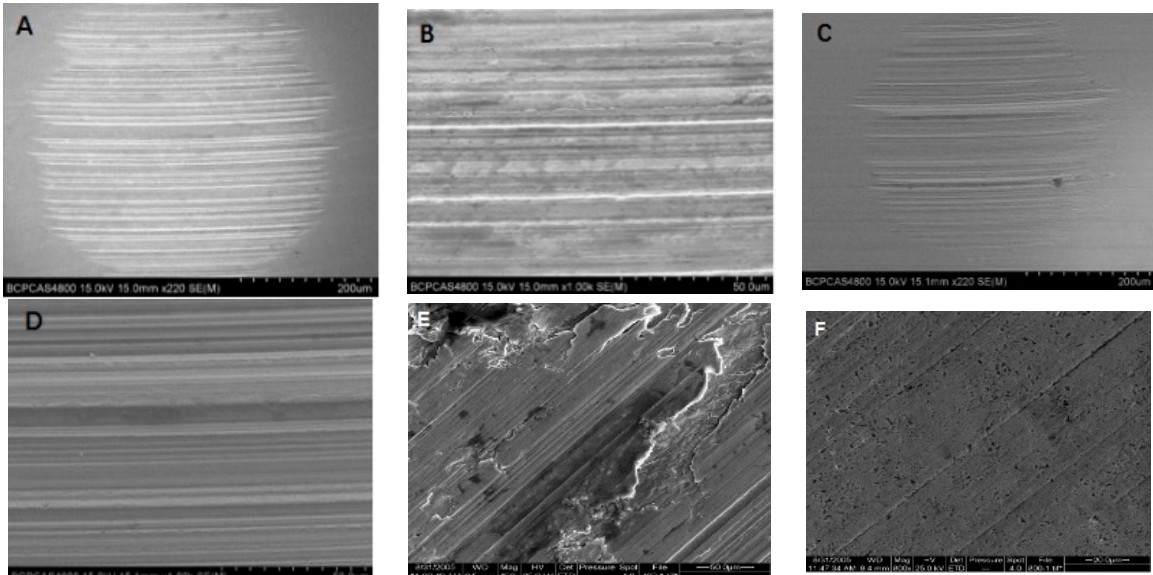

**Figure 5.** SEM images of wear tracks morphology [4]. (**A**,**B**,**E**)—without AMNP; (**C**,**D**,**F**)—with AMNP; (**A**–**D**) were carried out using an four-ball friction tester. (**E**,**F**) were carried out using a ring-on-disk tester. All friction tester are rotation speed of 1200 r/min, loading capacity of 392 N, and test duration of 60 min (**A**–**D**) and 480 min (**E**,**F**).

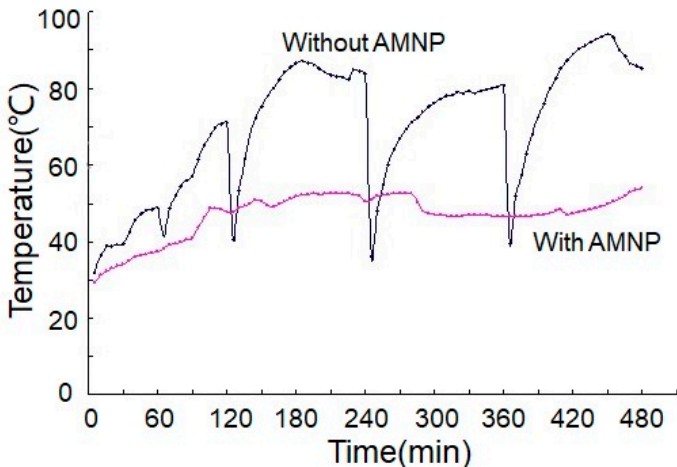

**Figure 6.** Changes of lubricating oil in friction test. (The figure is unpublished data by the corresponding author of this article).

Antigorite composite powders-oil system: a few studies have reported on the tribological performances of the composites of AMNP and other functional components (Table 1, No.40 to 46). For example, the composite powders containing AMNP and nano-metal compounds such as Mo, La, Cu and Ce, have a lower friction coefficient and wear volume than that of AMNP alone. It is believed that the addition of nano metal compounds is conducive to the formation of tribofilm through ion exchange, while rare earth elements (La and Ce) mainly act catalysts [19,54,55,64,66]. The AMNP (0.1%) + Zinc dialkyl dithiophosphate (ZDDP, 1.0%) has a lower wear volume of friction pair than that of AMNP alone (Figure 7). It is found that AMNP and ZDDP act together to form tribofilms containing AMNP and Zn polyphosphate, ZnS, from the decomposition of ZDDP on the surface of friction pair, which is a film that has a lower friction coefficient and wear. Polyphosphates are the essential ingredients of tribofilms and AMNP is an antiwear-additive [27].

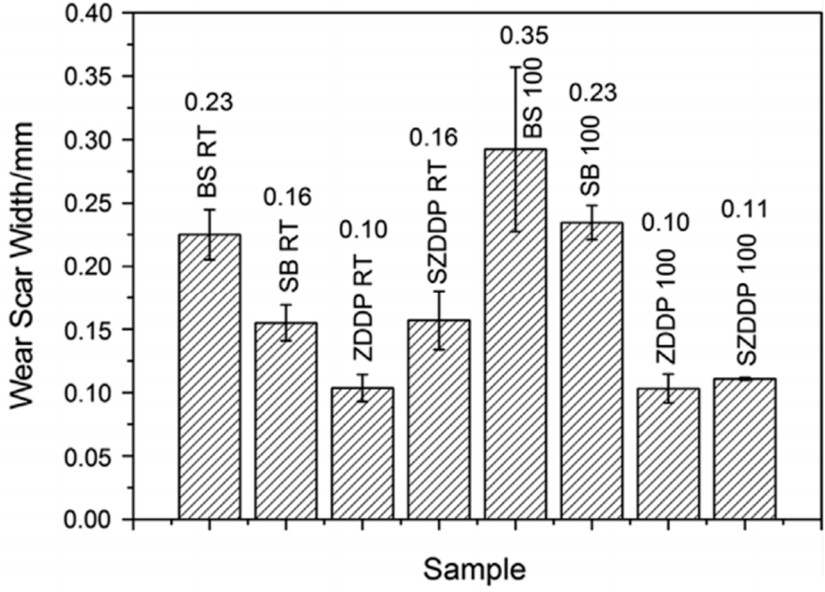

**Figure 7.** Comparison of wear scar width [27]. BS RT—without AMNP and ZDDP at 25 °C; SB RT—with AMNP at 25 °C ZDD RT with ZDDP at 25 °C; SZDDP RT—with AMNP and ZDDP at 25 °C BS 100—without AMNP and ZDDP at 100 °C; SB 100—with AMNP at 100 °C ZDDP 100—without ZDDP at 100 °C; SZDDP 100—with AMNP and ZDDP at 100 °C.

Solid composites containing AMNP: effective improvement of the tribological properties of polytetrafluoroethylene (PTFE) or Al matrix composites is the key to prolong their service life and performance. Some research works on the evaluation of tribological performances of composite materials made by mixing AMNP with polymer or metal-base alloys has been carried out in the last decade [50,62,68–71,73,88–90] (Table 1, No.47 to 66). Sleptsova et al., (2009) found that the friction coefficient and wear of PTFE with ≤5% AMNP decreased by 10–15% and 95.6–99.8% compared with those without AMNP, as well as their rupture strength, elongation at rupture and modulus of elasticity at rupture were similar to that without AMNP [68]. The investigators hypothesized that the key role in the wear resistance of the polymer composites with AMNP or vermiculite is played by the origin of the Mg contained in the mineral fillers. The Mg not only influences the character of the oxidation processes, which occur in accordance with the radical mechanism, but can also primarily affect the processes of structure formation [68]. TiAl and NiAl or SiAl alloys with ≤11% AMNP have lower friction coefficient and wear than those without AMNP at 25 to 800 °C (the friction coefficient and wear are reduced by 8–45.2% and 11.4–62.6% respectively). NiAl matrix composite filled with 8 wt.% AMNP has a dense and homogeneous microstructure, and good tribological properties [73]. Specifically, the friction coefficients and wear rates are fairly low when the sliding speeds are at a low level. As the sliding speed is increased to a high level, the friction coefficients and wear rates show a distinct increase. The findings indicate that the variations in friction coefficients and wear rates are similar with an increase in the applied load [73]. Moreover, SiAl alloys with AMNP have higher compression strength than those without AMNP. Because AMNP prevent the generation of dislocations on single or multiple slip planes and their pile up at the grain boundaries, the compressive strengths of AMNP-reinforced composite were better. It is found that SiAl alloys with AMNP have a thick and dense structured layer uniformly coated on the substrate, showing great anti-friction ability [71]. These results imply that AMNP has a promising application as an additive to PTFE and aluminum-based engineering materials.

AMNP-grease system: the grease is a semisolid lubricant. The majority of traditional greases are composed of petroleum and synthetic oils thickened with metal soaps and other agents such as clay, silica, carbon black and polytetrafluoroethylene [90]. In recent years, some researchers have added AMNP by 0.5–3.0% to a semisolid lubricant to evaluate its tribological performances [14,52,74–76]. The results showed that those types of grease with AMNP have higher critical load and welding load and have lower friction pairs wear (Table 1, No.67 to 74). For example, the critical load of grease with 1% AMNP is increased by 7.1–13% compared with that of without AMNP, welding load is increased by 12.8–25.7%, mean diameter of wear spots is reduced by 19.1–20.5% [14]. Those can be explained by AMNP stronger retention in the friction zone by the structural lubricating medium frame during the run-in period and the adhesion of the protective film to the substrate [14]. The higher critical load and welding load of grease with AMNP indicated that AMNP could be a good application prospect as anti-wear and extreme pressure additive for industrial lubricating grease.

Comparative study of antigorite and other minerals: aside from the evaluating the tribological properties of AMNP, the comparative study on tribological performances between AMNP and other silicate minerals such as talc, kaolinite, vermiculite, sepiolite, attapulgite has also been carried out. For example, the studies of Lyubimov et al. [14] and Yang et al. [75] indicate that grease containing AMNP or kaolinite or talc powders can reduce the friction coefficient and wear of friction pair, but the grease with kaolinite or talc has higher critical load, welding load and diameter of wear spots than that of with AMNP [14,75]. Friction coefficient and mass wear rate of PTFE with 2% AMNP or 2% vermiculite are lower than that of the without, but AMNP is more effective than vermiculite in reducing friction coefficient and wear, which may be related to the higher reactivity of AMNP than vermiculite. By comparing the tribological properties of AMNP, sepiolite and attapulgite powders dispersed in base oil, it is found that their friction coefficient and wear volume of friction pair are lower than that of base oil. It is interesting to note that the effect of sepiolite and attapulgite in reducing the friction coefficient and wear of friction pair is slightly better than that of AMNP. It is concluded that the better frictional performance of sepiolite and attapulgite under a high load may be related to their TOT

crystal structure (an octahedral layer (O) is sandwiched between two tetrahedral layers(T)), higher reactivity [91].

## 4. Application of Antigorite Lubricating Additive

The application of AMNP in industrial equipment has been studied mainly in locomotive and automobile engines, air compressors and gearing. The evaluation of the application effect mainly focused on the cylinder burst pressure of engine, fuel consumption of the engine, temperature of lubrication oil, CO and CH emissions of the engine, power consumption of the driving motor, vibration amplitude of gears, engine consumption of lubricating oils (Table 1, No.110 to 113). The major conclusion is that adding AMNP to traditional lubricating medium could improve cylinder pressure (increased by 2.7–11%) [85,86], reduce the consumption of engine fuel (by 2.5–7%), reduce consumption of lubricating oil (by 14.3–94.5%) [58,59,86,87], reduce the temperature of lubricating oil (9.7%) (Figure 8) [4] and reduce the emission of CO (39.5%) and CH compounds (29.5%) [86], as well as reduce power consumption of driving motor (9.12–13%) (Figure 9) [4,54,59]. Of importance is also that lubricating medium containing AMNP could obviously prolong the service life of the friction parts of the equipment and improve the service efficiency [83,87,92,93].

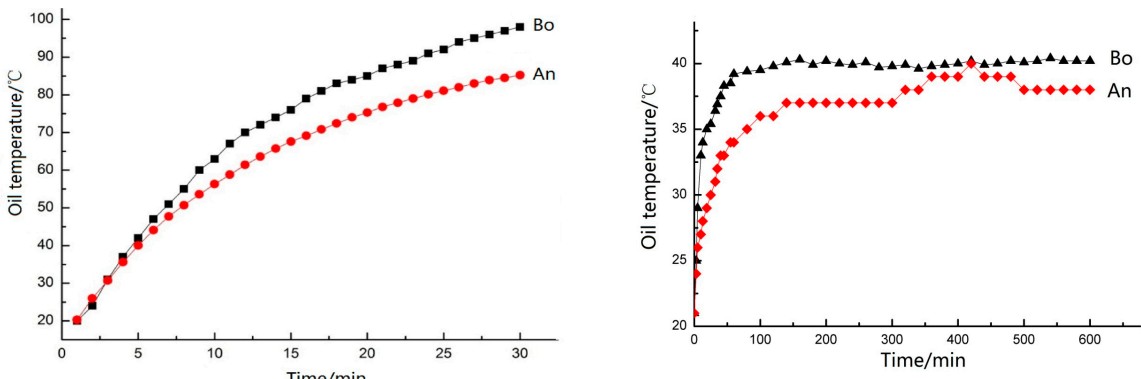

**Figure 8.** The change of temperature of lubrication oil over time in air compressor (**left**) and in gear (**right**) [4]. BO—without AMNP; An—with AMNP.

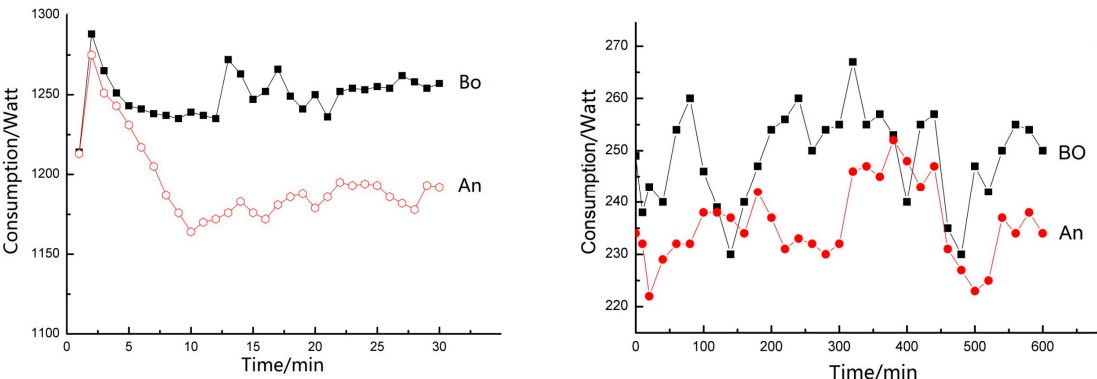

**Figure 9.** The change of power consumption of driving motor over time in air compressor (**left**) and in gear (**right**) [4]. BO—without AMNP; An—with AMNP.

## 5. Physico-Chemical Characteristics of a Friction Pair Surface

As already noted, AMNP as lubricant medium additive has participated in the friction reaction process. The study of chemical composition, phase composition, structure and physical properties of the surface layer of the friction pairs could reveal the essence of the interaction between the friction pair and the AMNP, which was the focus of tribology.

The chemical composition of friction pair surface layer is commonly studied using energy dispersive X-ray spectroscopy (EDS) coupled with scanning electron microscopy, X-ray photoelectron spectroscopy (XPS) and X-ray absorption near edge structure (XANES), while the phase composition and structure characteristics are usually observed through transmission electron microscopy (TEM) and atomic-force microscopy (AFM). Numerous studies have reported that the Si and Mg contents of most friction pairs treated with AMNP (Figure 10, middle image) were significantly higher than that of without AMNP [14,21–23,26,27,49–53,55,56,61,69,72,77,85,91,94–99]. Some friction pair surfaces treated with AMNP only detected Si but no Mg [17,25,54,55,64,66,74,78,79,84,86,91,95], or only detected Mg but no Si [18,60,100], or neither Si nor Mg were detected [57,82,83]. According to Zhao [101], the friction pair surface without AMNP mainly contains Fe, O and C, while in addition to Fe, O and C, Mg and Si appear on the surface with AMNP. Accordingly, Fe, O, C, Mg, Si, Zn, P and S appeared simultaneously on surface with AMNP + ZDDP (Figure 10). XANES spectra clearly show that the tribofilms produced by the lubricating oil adding AMNP and ZDDP have a glassy appearance containing antigorite and Zn polyphosphate, ZnS [27]. Of particular significance is the distribution of Si and/or Mg on the friction pair surface treated with AMNP is not uniform, and most of them are spots or patches rather than continuous layers, which is confirmed by TEM and XANES [26,27,77,84,95]. Zhao [101] showed that the tribofilm with maximum thickness of 240 nm and 458 nm are formed on the friction pair surface with AMNP and with AMNP + ZDDP (Figure 11). A number of studies have shown that the phase on the friction pair surface treated with AMNP is very complex, including nanocrystalline, amorphous and mineral cellular skeleton [26,27,77,84,95,101]. Figure 12A demonstrates a bright field transmission electron micrograph of the cross-section of the tribofilm, showing the thickness was about 500–600 nm. The film is uniform with a smooth surface and a few internal pores. The diffraction pattern shown in Figure 13B–D clearly indicates some crystallinity of the particles since diffraction rings are clearly visible. The rings were indexed and found to be the complex compounds: FeSi, AlFe and $Fe_3O_4$ complexes (Figure 12B); $Fe_3O_4$ and FeSi complexes (Figure 12C) and amorphous FeSi and $SiO_2$ complexes (Figure 12D) [77]. It can be inferred from the characteristics of composition and phase that the AMNP adsorbed or adhered to the friction surface forms an amorphous or nanocrystalline components structure layer with a discontinuous distribution under the influence of frictional dynamics and thermal energy.

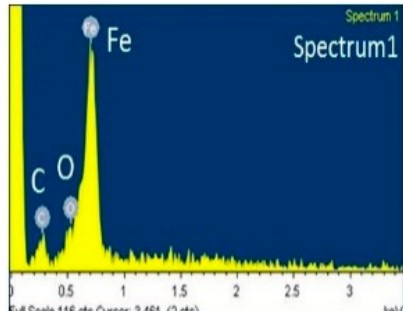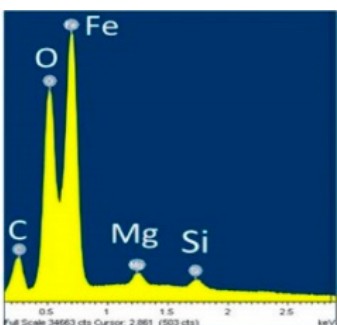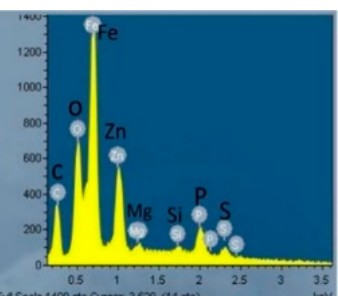

**Figure 10.** EDS analysis of friction pair surface [101]. Left—without AMNP; Center—with AMNP; Right—with AMNP and ZDDP.

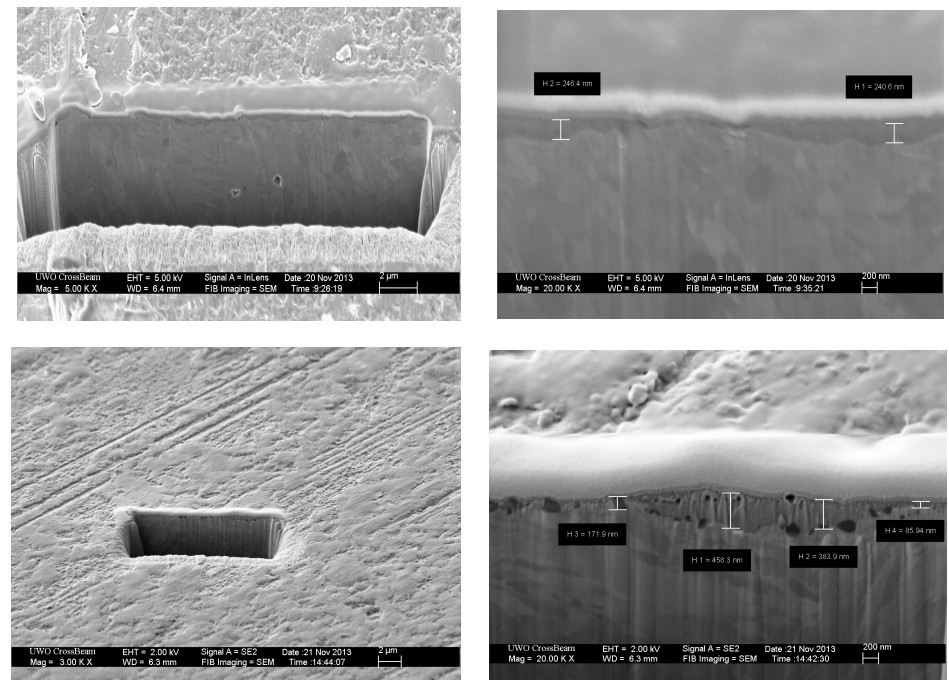

**Figure 11.** Focused Ion Beam/SEM images of the tribofilm [101]. Above—with AMNP; Below—with AMNP + ZDDP.

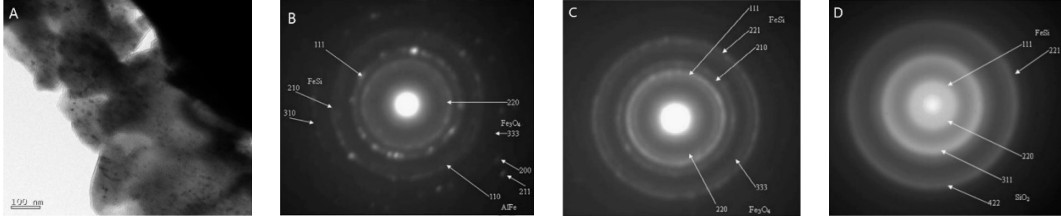

**Figure 12.** Transmission electron microscopy (TEM) images of the tribofilm with AMNP [77].

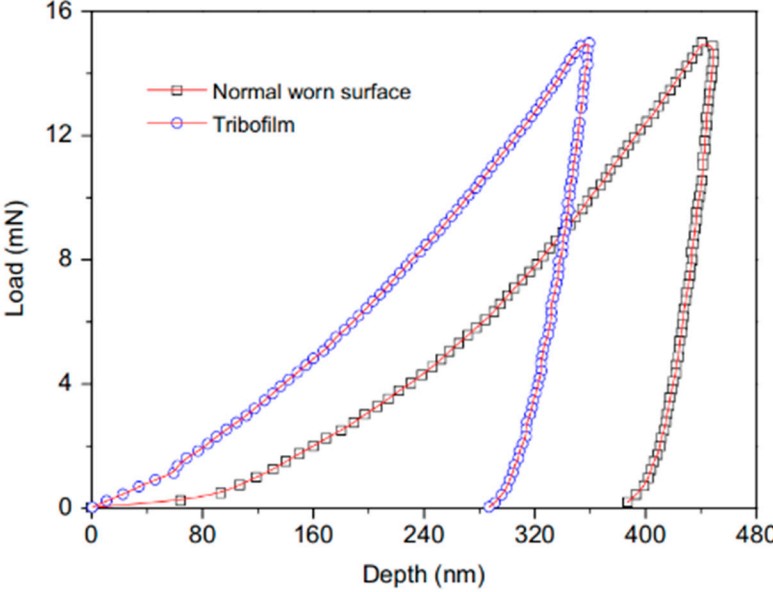

**Figure 13.** Typical load–depth curves of normal worn surface and the tribofilm [21].

In the comparison of the surface roughness (Ra) and hardness of the friction pair, we found that the roughness of the friction pair treated with AMNP is significantly lower than that without AMNP (Table 1, No.99–109; Ra of the friction pair with AMNP reduced by 40.2–72.1%), while the surface hardness is significantly higher (Table 1, No.75–88, the hardness increased by 1.8–94%).

Figure 13 and Table 2 show the typical load-depth curves, nano-hardness (H) and elastic modulus (E) of normal worn surface (without AMNP) and the tribofilm (with AMNP) tested with a maximal applied load of 15 mN. It is found that the maximum indentation depth for the tribofilm (with AMNP) is much smaller than those of normal worn surface (without AMNP). Correspondingly, tribofilm has a higher hardness and elastic modulus than normal worn surface, indicating the tribofilm was a hard–thin layer.

**Table 2.** Nano-mechanical properties of normal worn surface and the tribofilm [21].

|  | Max Depth (nm) | Plastic Depth (nm) | H (GPa) | E (GPa) | H/E ($\times 10^{-2}$) |
|---|---|---|---|---|---|
| Normal surface | 438.80 ± 59.70 | 393.50 ± 54.47 | 3.45 ± 0.85 | 215.53 ± 32.10 | 1.60 |
| Tribofilm | 342.55 ± 42.41 | 289.65 ± 37.57 | 6.68 ± 0.65 | 238.52 ± 29.65 | 2.80 |

The H/E ratio of nano-hardness (H) and elastic modulus (E) is usually introduced as a main parameter to estimate the relative wear resistance of materials. Figure 14 demonstrates H/E tested with different maximal indentation depth on a normal worn surface (without AMNP) and the tribofilm (with AMNP). The ratio varies between $1.61 \times 10^{-2}$ and $1.37 \times 10^{-2}$ for a normal worn surface with different maximal indentation depths, while it decreases from $3.41 \times 10^{-2}$ and $1.78 \times 10^{-2}$ for tribofilm with the increasing maximal indentation depth. The result indicates the wear resistance property of the tribofilm (with AMNP) is superior to the normal worn surface (without AMNP).

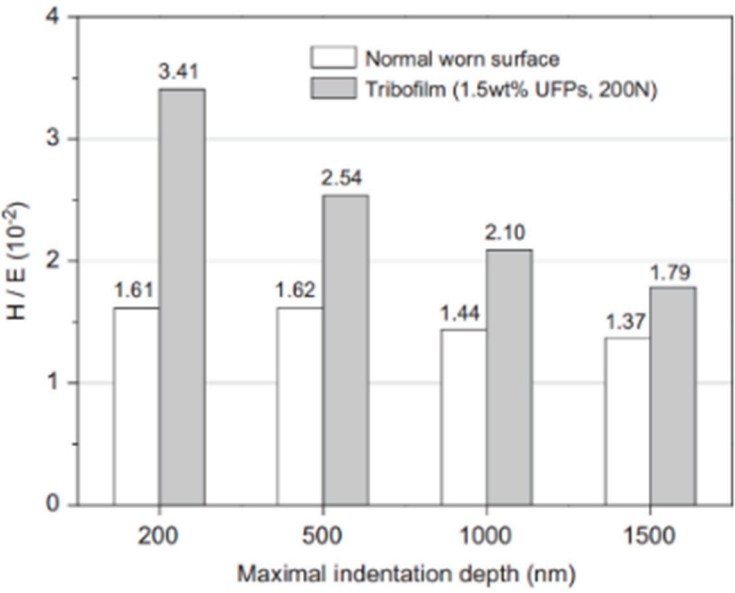

**Figure 14.** Dependency of H/E on maximal indentation depth [21].

Besides, it is interesting to note that the elastic modulus of the surface of the friction pair treated with AMNP does not show the same regularity as hardness and roughness (Table 1, No.89–98). In other words, the elastic modulus of the surface of the friction pair treated with AMNP is higher or lower than that of the one without AMNP.

## 6. Mechanism Study

A study on the lubrication mechanism of antigorite is of paramount importance in the understanding of the AMNP with friction pairs interaction, which can lead to the optimization of the subsequent preparation process and application technology of lubricity. The mechanistic studies have been carried out with either the assistance of surface composition and structure characterization of friction pairs, or the postulation from comprehensive experimental observations on antigorite mineralogy and friction characteristics. The internal relationship among antigorite mineralogy—surface features of friction pairs—application performance is shown in Figure 15.

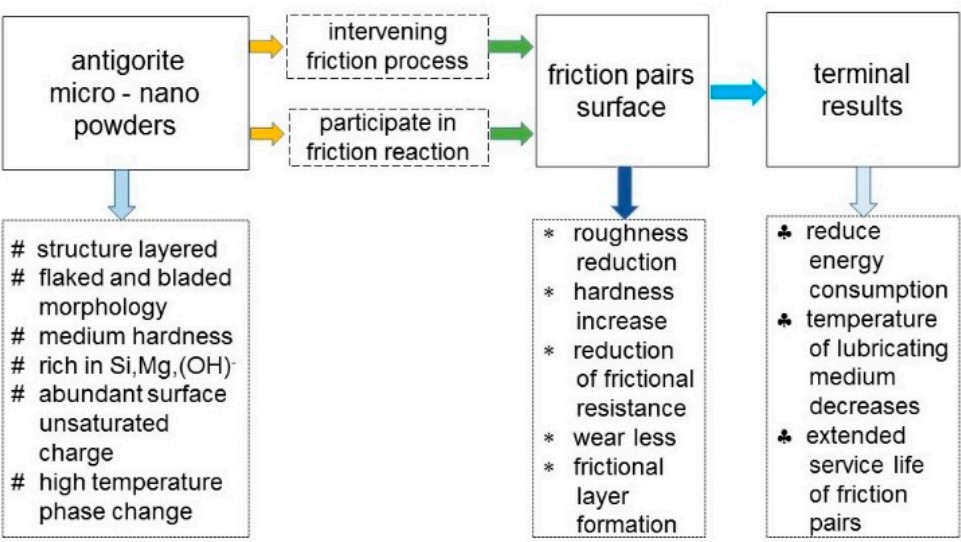

**Figure 15.** Schematic diagram of the relationship among antigorite mineralogy-friction pairs-application.

Pogodaev et al. [15] deduced from the structure, composition and phase transition that antigorite as a lamellar serpentine variety is more stable and durable to an external (mechanical) effect and to high temperature drops during its modification. Antigorite favors the formation of low dielectrically resistant, stronger, and wear-resistant protective layers on friction surfaces. In the presence of lubricating compositions with AMNP, the formation of superfinishing of friction surfaces may be due to the original availability of abrasive particles in powder like antigorite and to the abrasive effect of secondary particles present in the compositions after the AMNP decompose (forsterite, fayalite, oxides) [15].

Dolgopolov et al., found that the friction pair treated with AMNP has good wear resistance [16,17]. They thought that good wear resistance is due to the formation of a two-level structure on the rubbing surfaces. This structure is a mineral skeleton with a developed surface and a layer made up of the products of friction-polymerization of the lubricating materials [16]. The friction process involving AMNP is accompanied by substitution of magnesium atoms of the mineral structure by iron atoms from the friction surface. As a result, recombination of the crystalline lattice of the initial mineral and the formation of pseudo-mineral structures similar to the plane SiO compounds occur [17]. They found that antigorite is capable of forming a transfer film with tended crystals upon friction. It is assumed that under the friction of silicates, two substructures—silicate-oxygen and iron-oxygen—are formed in the transfer film [99].

Jin's research findings on generation mechanisms of reconditioning layer on cylinder bore of a locomotive diesel engine indicate that the strongly diffusing-in of oxygen atom, ion and free water from the metal surface by high chemical activity of antigorite results in oxidation of alloy component ($Fe_3C$) of Fe-based metal. This is a special internal oxidation process quite different from the high temperature internal oxidation which plays a crucial role in the formation of the reconditioning layer. Sequentially, the deformed refinement and strengthening of the internal oxidation structure

under reciprocating movement of the fiction pair bring about the reconditioning protective layer with excellent behaviors [102].

Bai et al. [4] thought that the flaked-bladed AMNP and their dehydrated products could serve as spacers and polishing media between asperities to eliminate direct metal-on-metal contact and adhesion, and polish the rubbing partners with rough surface. Abundant structural water provides interplanar lubrication and weakens antigorite's structure. Thermal phase transformation of antigorite and ion-exchange ($Fe^{2+}$ or $Fe^{3+}$ -$Mg^{2+}$) between dehydrated antigorite and iron base friction pairs induced the formation of ceramic-like film on the surface of the metals, which can greatly decrease friction and wear on friction surfaces [4].

Recently, Li et al., developed a comprehensive model to describe AMNP participating in the friction process and interaction of AMNP with friction pair (Figure 16) [44]. It is concluded on the basis of the summary of a large number of previous research results, which may be useful for understanding the mechanism of friction and wear of AMNP as an additive of lubricating medium. The model explains the mechanism of action of AMNP as a lubricating material from three aspects: the physical change of AMNP during the friction process, the chemical change during the interaction between AMNP with friction pair and the catalysis of antigorite to the formation of tribofilm. First of all, AMNP adsorbed on the surface of the friction pair is the basis for its particle size to gradually decrease and participate in the friction reaction so as to reduce the friction coefficient and wear of the friction pairs. Furthermore, the dehydrogenation and phase transition of antigorite under the action of friction heat and kinetic energy are the key factors for the surface oxidation of iron-base friction pair and the formation of composite gradient structure layer. In addition, the abundant surface unsaturated charge of antigorite powder and the catalytic effect of trace elements such as Ni and Mn on the interaction of lubricating oil—AMNP—friction pair are the motivating factors for the formation of tribofilm. The combined action of the above factors leads to the formation of tribofilm with high hardness, high elastic modulus and low friction coefficient and wear.

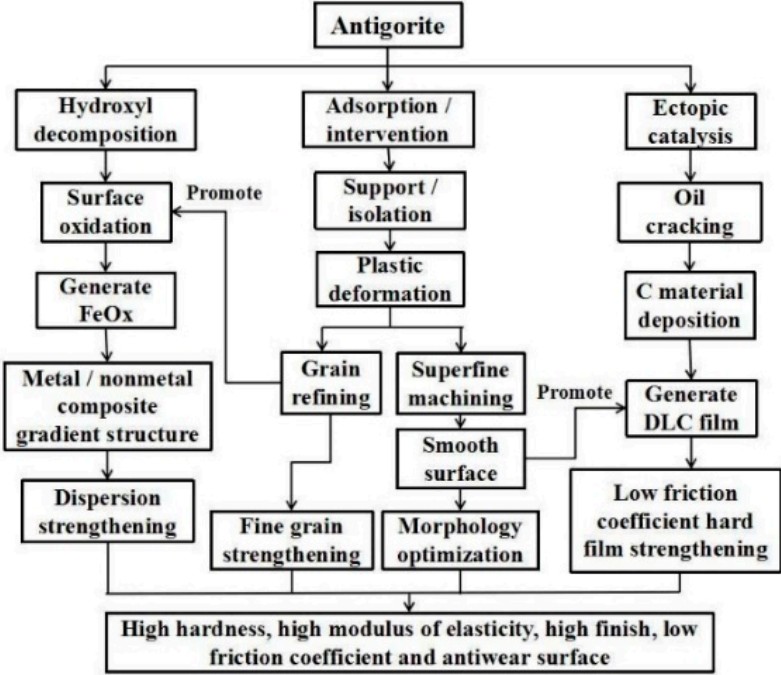

**Figure 16.** Schematic diagram of the antigorite powders–friction pair interaction [44].

## 7. Conclusions

Lubricity and anti-wear of antigorite may be related to its lamellar morphology, medium hardness, abundant hydroxyl water and surface unsaturated charge, strong ion exchange capacity, easy to slide

between layers and high temperature phase transition. Under the action of friction kinetic energy and heat energy, the bladed AMNP forms a friction reaction layer having low roughness and high hardness on the friction pair surface by abrasive polishing and ion exchange, which effectively reduces the friction resistance and wear of the friction pairs.

As a lubricant additive, AMNP should have a bladed morphology with a blade diameter less than 2 microns, and should be subject to surface modification of organic matter.

The AMNP can be mixed with different types of lubricating oil to form a suspended solution, or evenly mixed with base grease to make a complex grease, or mixed with matrix materials (such as PTFE and Al alloys) to make solid composite materials. AMNP used as a lubricant additive for PTFE can significantly reduce the wear of PTFE, which can extend the service life of the PTFE friction seal components.

As already noted, the tribological properties of calcined AMNP under 300 °C are better than that of the uncalcined AMNP. The AMNP used as a lubricating material should not be calcined at temperatures >600 °C.

The application of AMNP in industrial equipment could reduce friction wear and improve cylinder pressure; reduce the consumption of engine fuel and lubricating oil, as well as the temperature of lubricating medium and the emission of pollution, obviously prolong the service life of the friction parts of the equipment, extend the overhaul period of the equipment and improve the service efficiency.

Antigorite as a lubricant has not yet formed a set of standards and specifications, which is a technical obstacle that restricts the large-scale application of AMNP in industry.

In published papers, some authors indiscriminately refer to "antigorite" as "serpentine". In fact, the "serpentine" is a group of minerals that contains lizardite, antigorite and chrysotile [102]. Although the three minerals share the same composition ($Mg_6Si_4O_{10}(OH)_8$), they have different structures. Lizardite shows an ideal layer topology, whereas antigorite is a modulated layer and chrysotile is a bent layer. Of the three minerals, antigorite is currently the only additive lubricant, while fibrous chrysotile is not used because of its potential environmental hazards. The authors suggest that "antigorite" rather than "serpentine" should be used as a lubricant.

Some papers refer to antigorite as a "self-repairing" material based on its anti-wear properties [23,44,73]. The word "self-repairing" means that the object itself has a self-repairing function. In fact, antigorite was not repaired during the friction process, but the friction pair using AMNP was repaired. So, this paper argues that "self-repairing" may be an inappropriate term to describe the effects and functions of antigorite, and so should be abandoned.

**Funding:** This research was funded by National Key R&D Program of China(2017YFB0310703).

**Acknowledgments:** This project was sponsored by the National Key R&D Program of China(2017YFB0310703).

**Conflicts of Interest:** The authors declare no conflict of interest.

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
