# Peer review of "Tribological Performance and Application of Antigorite as Lubrication Materials"

_lubricants, doi:10.3390/lubricants8100093_

Round 1

Reviewer 1 Report

The authors have made a number of improvements to their manuscript since their previous version. Additional figures have been included and discussion is now more detailed

I recommend that further revision of this manuscript is necessary. These amendments are listed below.

  1. Please include references in the figure captions. They are missing from all the figures. In many cases, relevant information is also lacking from the figure captions.
  2. Section 2 (line 66): please include units in the hardness measurements (e.g. HV), including the load used to obtain the hardness, if known.
  3. Section 2(line 67): what is the reason for the difference in theoretical density and measured density?
  4. Figure 2 needs more explanation, in particular, what feature denotes dehydration and which one indicates dihydroxylation. Is the dihydroxylation the change in slope of the TG curve between 600 and 800°C? What are the various peaks in the DTA curve?
  5. Section 2 (line 77): please amend to “…morphology of natural antigorite…”
  6. Section 2 (line 84): please amend to “…pH of approximately 10…”
  7. Section 2 (line 87): what are Span60 and Tween60?
  8. Figure 6: what test and what test conditions were used on the specimens shown in this figure?
  9. Figure 7: the axis labels are hard to read in this figure.
  10. Section 3 (line 177): please amend to “Sleptsova et al [add ref no.] found that…” (i.e. removed the words “research”.
  11. Section 3 (line 219): please amend to “Comparison of antigorite and other minerals”.
  12. Figure 14: Typical load-depth curves of what?
  13. Section 5 (line 304-307): what are the respective hardness and modulus values of the two load-depth curves shown in Figure 14?
  14. Section 6 (line 325): please amend to “…and anti-wear behaviour of…”
  15. Section 6 (line 342) – “Dolgopolov et al found that…” – reference?
  16. Section 6 (line 345): please amend to “…tribopolymerization…”
  17. Section 7 (line 387): please amend to “Lubricity and anti-wear behaviour…”

Author Response

Response to Reviewer #1:

  1. Please include references in the figure captions. They are missing from all the figures. In many cases, relevant information is also lacking from the figure captions.

The corresponding to the problem: In addition to the newly compiled chart in this paper, other attached charts have been annotated with references.

  1. Section 2 (line 66): please include units in the hardness measurements (e.g. HV), including the load used to obtain the hardness, if known.

The corresponding to the problem: Vickers hardness is dimensionless hardness, which is calculated based on indentation length and load.

  1. Section 2(line 67): what is the reason for the difference in theoretical density and measured density?

The corresponding to the problem: Theoretical density is calculated according to cell parameters, while actual density is determined according to Archimedes principle.There are often differences between the two.

  1. Figure 2 needs more explanation, in particular, what feature denotes dehydration and which one indicates dihydroxylation. Is the dihydroxylation the change in slope of the TG curve between 600 and 800°C? What are the various peaks in the DTA curve?

    The corresponding to the problem: Figure 2 has been explained in detail in the article.

  1. Section 2 (line 77): please amend to “…morphology of natural antigorite…”

The corresponding to the problem: It has been modified to“nature antigorite is commonly bladed or fibrous.

  1. Section 2 (line 84): please amend to “…pH of approximately 10…”

The corresponding to the problem: It has been modified in the article.

  1. Section 2 (line 87): what are Span60 and Tween60?

The corresponding to the problem: The full English name of Span60 and Tween60 have been added to the text.

  1. Figure 6: what test and what test conditions were used on the specimens shown in this figure?

The corresponding to the problem: The test conditions and apparatus have been supplemented in the diagram.

  1. Figure 7: the axis labels are hard to read in this figure.

The corresponding to the problem: We have redrawn Figure 7.

  1. Section 3 (line 177): please amend to “Sleptsova et al [add ref no.] found that…” (i.e. removed the words “research”.

The corresponding to the problem: We had removed the words “research” and added ref no.

  1. Section 3 (line 219): please amend to “Comparison of antigorite and other minerals”.

The corresponding to the problem: We've already made the changes: Comparative study of antigorite and other minerals

  1. Figure 14: Typical load-depth curves of what?

The corresponding to the problem: We've already made the changes: Fig. 14 Typical load–depth curves of normal worn surface and the tribofilm [21]

  1. Section 5 (line 304-307): what are the respective hardness and modulus values of the two load-depth curves shown in Figure 14?

The corresponding to the problem: We have supplemented the corresponding nano hardness and elastic modulus in Table 2.

  1. Section 6 (line 325): please amend to “…and anti-wear behaviour of…”

The corresponding to the problem: We have modified the relevant sentences.

  1. Section 6 (line 342) – “Dolgopolov et al found that…” – reference?

The corresponding to the problem: We had added ref no.

  1. Section 6 (line 345): please amend to “…tribopolymerization…”

The corresponding to the problem: tribopolymerization has been modified.  

  1. Section 7 (line 387): please amend to “Lubricity and anti-wear behaviour…”

The corresponding to the problem: “Lubricity and anti-wear behaviour has been modified

Reviewer 2 Report

The authors have potentially improved the manuscript. They answered the raised questions, included more discussion and illustrative/informative figures.

I think the manuscript can be accepted for publication after taking into account the following:

  • Obtaining copyright from the original authors/publishers of Figures 2, 3, 5-15.
  • Aligning values and references in tables as they appear to in random orientation.

Author Response

Response to Reviewer #2:

  • 1)Obtaining copyright from the original authors/publishers of Figures 2, 3, 5-15.
  • The corresponding to the problem: The Figures used in this paper are all provided by the author and the references are noted.
  • 2) Aligning values and references in tables as they appear to in random orientation.

The corresponding to the problem: The technical treatment for Table 1 has been done.

Round 2

Reviewer 1 Report

This manuscript is greatly improved following the recent revisions by the authors. However, it still requires further revisions before it can be accepted for publication.  Requested changes are listed below.

Line 71: typographical error. This should read “The first apparent exothermic peak at 793 °C partially overlaps…”

Line 81: should read “Natural antigorite…”

Line 282/283: typographical error. I think this should read “…Mg and Si appear on the surface with AMNP”. This would then be consistent with the middle image in Figure 11, which, according to the caption, shows an EDS spectrum “with AMNP”.

Several of the references given in the figure captions are incorrect. These are listed below.

Figure 7 is not in reference [4].

Figure 8 is from reference [21] and not [27]. In addition, the discussion of Figure 8 in the text is a comparison of the wear volumes of a friction pair when AMNP+ZDDP is used with that of AMNP alone. The image, which is exactly the same as Figure 15, is clearly the wrong image for Figure 8.

Figure 11 is not from reference [4].

Figure 12 is not from reference [4].

Figure 15 is exactly the same as Figure 8 (see above), although, from the discussion in the text, appears to be the correct image for Figure 15.

Figure 17 is not from reference [4].

Author Response

Dear reviewer and editor , Thank you very much for your helpful advice on our paper '' Tribological performance and application of antigorite as lubrication materials ''. The reviewers' comments and suggestions were highly insightful and enabled us to greatly improve the quality of our manuscript. We performed a careful proof-reading of the manuscript to minimize errors.

Response to Reviewer #1:

  • Line 71: typographical error. This should read “The first apparentexothermic peak at 793 °C partially overlaps…”

The corresponding to the problem: It has been modified according to the suggestions of reviewers.

2) Line 81: should read “Natural antigorite…”

The corresponding to the problem: It has been modified according to the suggestions of reviewers.

3)Line 282/283: typographical error. I think this should read “…Mg and Si appear on the surface with AMNP”. This would then be consistent with the middle image in Figure 11, which, according to the caption, shows an EDS spectrum “with AMNP”.

The corresponding to the problem: It has been modified according to the suggestions of reviewers, and references have been added in Figure 11.

4)Several of the references given in the figure captions are incorrect. These are listed below.

(1)Figure 7 is not in reference [4].

The corresponding to the problem: Figure 7 is the unpublished data of the corresponding author of this article, which has been explained in the notes

(2) Figure 8 is from reference [21] and not [27].

The corresponding to the problem: It has been modified according to the suggestions of reviewers.

(3) In addition, the discussion of Figure 8 in the text is a comparison of the wear volumes of a friction pair when AMNP+ZDDP is used with that of AMNP alone. The image, which is exactly the same as Figure 15, is clearly the wrong image for Figure 8.

The corresponding to the problem: Sorry, there is a paste error in this figure, we have done the substitution.

(4)Figure 11 is not from reference [4].

The corresponding to the problem: We have changed literature [4] into literature [111].

(5) Figure 12 is not from reference [4].

The corresponding to the problem: We have changed literature [4] into literature [111].

(6)Figure 15 is exactly the same as Figure 8 (see above), although, from the discussion in the text, appears to be the correct image for Figure 15.

The corresponding to the problem: We have replaced Figure 8.

(7) Figure 17 is not from reference [4].

The corresponding to the problem: We have changed literature [4] into literature [44].

This manuscript is a resubmission of an earlier submission. The following is a list of the peer review reports and author responses from that submission.

Round 1

Reviewer 1 Report

I would like to acknowledge the efforts that the authors have made to improve the manuscript since submission of the previous version. As well as adding more references, the authors have expanded the discussion of most sections of the manuscript and provided more detail. However, for a review paper, the level of detail in the discussion of various studies in the literature is no more than superficial. Also, the figures are still not explained in the text and do not add anything to the discussion. A selection of carefully chosen figures would greatly enhance the quality of the paper. Some suggestions for possible figures include: (i) a schematic diagram of the antigorite crystal structure; (ii) micrographs of worn surfaces in tribo-systems containing antigorite additives, including those of tribo-films; (iii) graphs of friction coefficient and/or wear rate comparing tribo-systems with and without antigorite; (iv) micrograph of antigorite powders; (v) schematic diagram of the tribo-film formed by antigorite; (vi) other characterisation data of antigorite tribo-films (for example XRD, EDS, nanoindentation load-depth curves etc.).

The authors also need to discuss the various studies in more detail and the discussion should be more than a comparison of friction coefficients and wear. The manuscript would also benefit from a description of the nature and properties of surface tribo-films formed in systems in which antigorite additives are present. As an example, nanoindentation has been used to probe the hardness and elastic modulus of tribo-films (see, for example, Yu et al., Tribol. Int., 43 (2010), 667-675): the authors might like to compare the properties of antigorite-based tribo-films with those of other anti-wear additives such as ZDDP. However, in its present form, the manuscript contains almost no discussion of the properties of such films.

In summary, this manuscript needs major revision before it can be considered to be suitable for publication. Before they make their revisions, I suggest the authors do an internet search on “what makes a good review article” for information on how to write their review.

Reviewer 2 Report

Journal Lubricants (ISSN 2075-4442)

Manuscript ID   lubricants--886727

The manuscript entitled “Tribological performance and application of antigorite as lubrication materials,” prepared by Bai and coworkers is a modified version of a previous submission.

In this manuscript, the authors provided additional data, explanation, and references than the previous one. The manuscript is more comprehensible. However, some points need to be considered before this manuscript becomes acceptable for publication:

Page1, line 38: introduce briefly what is meant by functional repairing materials.

Page 2, line 65: the phrase between parentheses is rather confusing than comprehensive, better to rephrase it or to remove it “(hydroxyl water formation from OH groups and its removal)”

Table 1: justification is required on the distinct performance of materials having similar composition reported by different groups: one example is the entries 4, 7, and 9. Why is there a difference? Citing the reference is not enough. Readers expect to find the answer in the review. Is the difference due to type of oil used? preparation procedure? uncertainty of measurement?...

Table 1: “The surface composition and phase of the friction pair with AMNP” section does not bring any value to the review. The same information are available in Figure 1, or discussed in the

“Physico-Chemical Characteritics of Friction Pair Surface” section.

Figure 3. is not cited in the text.  Authors are requested to explain and cite this figure, or to remove it

Proof reading is required to correct many format, spelling, grammar, and punctuation mistakes.

  • Several typing, punctuation, and grammar errors are present at different parts of the text.
  • Reference citation: all reference citations in table 1 should be aligned
  • Spaces separating the text from reference citation is missing in many places (some examples: page 2 line 70, line 77, page 3 line 115, line 120).